# Correlation of Computed Tomography Parameters with Histology, Stage and Prognosis in Surgically Treated Thymomas

**DOI:** 10.3390/medicina57010010

**Published:** 2020-12-24

**Authors:** Angelo Carretta, Stefano Viscardi, Piergiorgio Muriana, Paola Ciriaco, Alessandro Bandiera, Roberto Varagona, Michele Colombo, Giampiero Negri

**Affiliations:** 1Department of Thoracic Surgery, San Raffaele Hospital, Via Olgettina, 60-20132 Milan, Italy; viscardi.stefano@hsr.it (S.V.); muriana.piergiorgio@hsr.it (P.M.); ciriaco.paola@hsr.it (P.C.); bandiera.alessandro@hsr.it (A.B.); negri.giampiero@hsr.it (G.N.); 2School of Medicine, Vita-Salute San Raffaele University, 60-20132 Milan, Italy; 3Department of Radiology, San Raffaele Hospital, Milan, Via Olgettina, 60-20132 Milan, Italy; varagona.roberto@hsr.it (R.V.); colombo.michele@hsr.it (M.C.)

**Keywords:** thymoma, surgery, histology, staging, CT, prognosis

## Abstract

*Background and objectives*: The histological classification and staging of thymic tumors remains a matter of debate. The correlation of computed tomography (CT) parameters with tumor histology and stage also still has to be completely assessed. The aim of this study was therefore to analyze the correlation of radiological parameters with histological and staging classifications of thymomas evaluating their prognostic role. *Materials and Methods*: Data of 50 patients with thymoma submitted to a complete surgical treatment between 2005 and 2015 were retrospectively analyzed. Tumors were classified according to the WHO and Suster and Moran (S&M) histological classifications and to the Masaoka–Koga and tumor, node and metastases (TNM) staging systems. The correlation of CT features with histology and stage and the prognostic role of histopathological and radiological features were assessed. *Results*: Five-year overall (OS) and disease-free survival (DFS) were 90.3% and 81.1%, respectively. A significant correlation of DFS with the Masaoka–Koga (*p* = 0.001) and TNM staging systems (*p* = 0.002) and with the S&M (*p* = 0.02) and WHO histological classifications (*p* = 0.04) was observed. CT scan features correlated with tumor stage, histology and prognosis. Moderately differentiated tumors (WHO B3) had a significantly higher incidence of irregular shape and contours (*p* = 0.002 and *p* = 0.001, respectively) and pericardial contact (*p* = 0.036). A larger tumor volume (*p* = 0.03) and a greater length of pleural contact (*p* = 0.04) adversely influenced DFS. The presence of pleural (*p* < 0.001) or lung invasion (*p* = 0.02) and of pleural effusion (*p* = 0.004) was associated with a significantly worse OS. *Conclusions*: Pre-operative CT scan parameters correlate with stage and histology, and have a prognostic role in surgically treated thymomas.

## 1. Introduction

Thymomas are rare tumors, accounting for less than 1% of all adult neoplasms, with an incidence of 1.7 per million person-years in Europe [1]. Complete surgical resection is the mainstay of treatment of these tumors, associated with induction and adjuvant treatments in more advanced stages of the disease [2]. The therapeutic strategy depends not only on tumor stage but also on the histological characteristics. However, at present no unique consensus exists concerning the optimal histological and staging classification of these tumors.

The World Health Organization (WHO) [3,4] histological classification of thymic tumors is the most frequently used, but due to limitations in terms of reproducibility, other classifications, such as the one described by Suster and Moran (S&M), have been proposed [5]. Staging of thymic tumors has been traditionally performed by means of the Masaoka–Koga (M-K) system [6]. However, in more recent years a tumor, node and metastases (TNM) classification has also been introduced in clinical practice [7]. The potential advantage of the TNM staging is a better prognostic evaluation in the preoperative setting, since the Masaoka–Koga is by definition a pathological classification. However, re-classifying the tumors according to the TNM staging system is associated with a stage migration effect, with a shift towards TNM stage I of tumors previously classified as M-K stage I, II, and even III. These adjustments, along with the concomitant use of both classifications, may lead to uncertainty in identifying a correct and univocal course of treatment. Other preoperative parameters are therefore required to define the therapeutic strategy of thymic tumors. In particular, the correlation between radiological features and histo-pathological parameters could allow us to improve the preoperative definition of the surgical strategy. In fact, the role of computed tomography (CT) scan parameters in the preoperative assessment of stage and histology of thymomas has been analyzed only in few previous trials [8,9]. The identification of preoperative CT scan parameters correlated with histo-pathological features could allow a prognostic stratification of thymomas and influence the surgical strategy, in particular concerning the extent of tumor resection and lymphoadenectomy.

The aim of this retrospective study is therefore to evaluate the correlation between preoperative CT scan features and histological and pathological characteristics of thymomas, assessing their prognostic role.

## 2. Materials and Methods

### 2.1. Study Design

Following informed consent for data collection, patients submitted to a complete surgical treatment for pathologically proven thymoma at the Department of Thoracic Surgery of the San Raffaele Scientific Institute in Milan, Italy between January 2005 and December 2015 were retrospectively reviewed. Exclusion criteria were a histological diagnosis of thymic carcinoma and, since disease-free survival (DFS) was defined as a primary endpoint of the study, an incomplete tumor resection (R1 or R2).

The following data were entered in a database: CT scan parameters, age, gender, associated myasthenia gravis, induction and adjuvant treatments, date of operation, surgical approach (median sternotomy, thoracotomy or VATS (Video-Assisted Thoracic Surgery)). Tumors were classified according to the 2015 WHO and to the Suster and Moran histological classifications, and staged according to the Masaoka–Koga and TNM staging systems. Disease-free (DFS), overall (OS) survival and cause of death were assessed.

Postoperative follow-up was performed with CT scan every 3 months for the first 3 years after surgery, then every 6 months for the next 2 years, and annually thereafter.

This retrospective analysis was approved by the local Institutional Review Board and has been performed in accordance with the ethical standards of the Declaration of Helsinki. The trial has been registered on clinicaltrials.gov (NCT04577495).

### 2.2. Histopathologic and Staging Review

All histopathological reports were reviewed. Histology of each tumor was classified according to the 2015 WHO classification and re-classified according to the classification proposed by Suster and Moran as follows: A, AB, B1 and B2 thymomas were re-classified as well-differentiated thymic neoplasms (or typical thymoma); B3 tumors were re-classified as moderately differentiated thymic neoplasms (or atypical thymoma).

Tumors were staged according both to the Masaoka–Koga and the 8th edition of the IASLC/ITMIG TNM staging system. Lymph node dissection was not routinely performed in the period analyzed. When no lymph nodes were identified in the surgical specimen the tumor was classified as Nx.

### 2.3. CT Scan Features and Image Interpretation

All patients underwent chest CT scan before surgery. Images were acquired with two scanners: a 16-row multidetector CT (MDCT) (Toshiba Aquilion, Otawara, Tochigi, Japan) and a 64-row MDCT (Philips Briliance 64 s, Cleveland, OH, USA). A first acquisition without contrast material injection was obtained, with a second acquisition 60 s after intra-venous injection of contrast material (*iopromide*, with iodine concentration of 370 mg/mL) in a single breath hold at the end of inspiration. Technical parameters were set at 120 kVp, 180 mAs, pitch of 1, section thickness of 1 mm, contiguous section interval, and 512 × 512 matrix. Observation was performed on soft tissue window.

CT scans were reviewed by two radiologists, expert in thoracic oncology (MC, RV) blinded to the histopathological diagnosis. Differences were resolved by consensus. The following features were evaluated using a dedicated software (Philips Intellispace, Philips Healthcare, Best, The Netherlands) and are reported in Appendix A: location of the tumor, 2-axes diameters, volume, shape, presence of necrosis, calcifications, pathological lymph nodes, presence and length of pleural contact, presence of pleural effusion or dissemination, pericardial effusion, invasion of mediastinal fat, great vessels, pericardium or lung, and contrast-enhancement pattern.

Location of the tumor was classified as right, left, or median according to the site of the intersection of the two main diameters of the lesion. The size of the lesion was defined as the length of the largest of the three lesion diameters. Tumor volume was calculated with the dedicated software after the identification of the borders of the lesion. Shape and contours were defined as regular or irregular. A contrast-enhancement pattern was assessed in the images obtained 30 s after contrast medium injection, and described as homogeneous or heterogeneous. Necrosis was defined as an area without contrast-enhancement or low-density value comparable to water. A short-axis diameter of ≥1 cm was used as the threshold for pathological lymph-nodes. Length of pleural contact was measured with multiple lines along all the borders of the lesion in contact with the mediastinal pleura. Invasion of the mediastinal fat was established in case of disomogeneity, ill-definition and hypodensity. Contact with great vessels was reported whenever a clear distinction of margins was not possible. Lung invasion was documented when the border between the tumor and the lung was markedly irregular and/or the lung was compressed and pinched by the lesion. Final diagnosis was reached by agreement between the two radiologists.

### 2.4. Statistical Analysis

Analysis was performed with the SPSS software v. 18 (SPSS Inc., Chicago, IL, USA). In order to evaluate the association between clinical variables, histology, radiological features and prognosis, continuous variables were dichotomized according to their median value. Comparisons of categorical variables among the groups of patients were performed by means of either Chi-square test or Fisher Exact test when appropriate. Survival curves were estimated with the Kaplan and Meier method. Cox regression analysis was used to assess the risks of the variables. Survival rates of patients grouped according to selected variables were compared by means of the log-rank test. Disease-free survival (DFS) was identified as primary endpoint. A multivariate analysis was performed using the Cox regression method to evaluate the independent contribution of the variables to recurrence-free and overall survival. Hazard Ratio (HR) and 95% Confidence Interval (95% CI) are shown; a *p*-value < 0.05 was considered statistically significant.

## 3. Results

### 3.1. General Clinical Features

Between 2005 and 2015, 50 patients submitted to a complete surgical treatment for pathologically proven thymoma were identified. The follow-up was complete for all the patients. The characteristics of the patients included in the study are summarized in Table 1.

The median follow-up was 57.5 months (range 3–182 months). At the completion of the study, 35 patients were alive with no evidence of disease, 8 were alive with disease recurrence, 1 died of disease recurrence and 6 died of other causes.

One-, 5- and 10-year overall survival was 100%, 90.3% and 84.3%, respectively. Disease-free survival at 1, 5 and 10 years was 97.9%, 81.1% and 56.5%, respectively.

### 3.2. Correlation of Histology and Staging

Thirty-seven WHO type A to B2 thymomas (74%) were re-classified as typical thymomas according to the Suster and Moran classification; 13 WHO type B3 thymomas (26%) were re-classified as moderately differentiated tumors. Eight out of 13 patients (62%) with a moderately differentiated (WHO B3) thymoma had myasthenia gravis (*p* = 0.002).

All Masaoka–Koga stage I and II tumors, and one stage III tumor (78%) were re-classified in stage I according to the TNM staging system (Table 2). Thirty-five out of 37 well-differentiated thymomas (WHO A to B2) (95%) were classified in stage I and 6 out of 13 moderately differentiated tumors (WHO B3) (46%) were classified as stage III thymomas according to TNM (Table 3). Univariate analysis showed a significant correlation of the Suster and Moran histological classifications with the TNM (*p* < 0.001) and the Masaoka–Koga (*p* < 0.001) staging systems, and of the WHO histological classification with the TNM (*p* = 0.009) and the Masaoka–Koga staging systems (*p* < 0.001).

### 3.3. Correlation of CT Scan Parameters, Histology and Staging

A significant correlation between CT scan parameters and the WHO and Suster and Moran histological classifications was observed. In particular, moderately differentiated tumors (WHO B3) had a significantly higher incidence of irregular shape and contours (*p* = 0.002 and *p* = 0.001, respectively), mediastinal fat invasion (*p* = 0.05) and pericardial contact (*p* = 0.036) (Figure 1 and Figure 2), (Appendix A).

A higher incidence of regular contours (*p* = 0.034) and a lower incidence of lung (*p* = 0.009) invasion were observed in patients with stage I tumors according to the TNM staging system (Figure 3). Stage I tumors according to M-K had a higher incidence of oval shape (*p* = 0.003) and a lower incidence of mediastinal fat invasion (*p* = 0.022), contact with mediastinal vessels (*p* = 0.008) and pericardium (*p* = 0.032) (Table 4).

### 3.4. Correlation with Survival

A significant correlation of the TNM (*p* = 0.002) and Masaoka–Koga (*p* = 0.001) (Figure 4) staging systems with DFS was observed. Furthermore, the T (*p* < 0.001) descriptor significantly correlated with DFS, while the N descriptor did not (*p* = 0.86). A sub-analysis of stage I showed a significant difference in DFS of T1a and T1b tumors (5-year survival 96.2% vs. 50%, *p* = 0.001). A significant correlation of DFS with the Suster and Moran (*p* = 0.02) (Figure 5) and WHO classifications (*p* = 0.04) was also observed.

At univariate analysis, a significant correlation of the WHO histological classification and overall survival was observed (*p* = 0.04). Conversely, no correlation of OS with the Suster and Moran histological classification (*p* = 0.08), the Masaoka–Koga (*p* = 0.40) and TNM (*p* = 0.83) staging systems was demonstrated.

Preoperative CT scan features correlated with DFS. In particular, a larger tumor volume (*p* = 0.03) (Figure 6) and a greater length of pleural contact (*p* = 0.04) (Figure 7) adversely influenced DFS. Moreover, the presence of pleural (*p* < 0.001) and lung invasion (*p* = 0.02), and of pleural effusion (*p* = 0.004) were associated with a significantly worse OS.

## 4. Discussion

Thymic epithelial tumors are rare and indolent neoplasms. Although long-term survival may be observed after complete surgical treatment, up to 50% of the tumors recur during follow-up [10]. Hence, several studies tried to identify pathological [11], radiological [9] and bio-humoral [12] features able to predict the risk of relapse after surgical treatment. Differentiated approaches in terms of extent of surgical resection and nodal dissection according to tumor stage and histology have been suggested [13], and therefore an accurate preoperative assessment is mandatory to correctly define the surgical strategy.

The histological classification of thymomas has been a matter of discussion for long time. The WHO classification of thymic tumors is based on the morphology of the prevalent type of cells of the tumor: spindle/oval cells (type “A”), round/epithelioid (type “B”, additionally subclassified as B1, B2 and B3 according to the proportional increase in cellular atypia) or their combination (type “AB”). In the WHO classification a linear progression of malignancy from A type to B3 was postulated [14,15], with a correspondent progressive increase in the relapsing risk, data confirmed in the analysis of our cohort. On the other hand, other studies failed in confirming the reproducibility of the WHO classification [16,17]. These discrepancies may be justified by a misinterpretation of cyto-histological parameters due to the complexity of the classification system, particularly in thymic tumors characterized by morphological heterogeneity [18]. Reducing complexity and overlaps in a classification system could allow an increase in the reproducibility and uniformity of the results [16] with a significant impact on therapeutic results. Therefore, Suster and Moran proposed a histological classification with two tumors subgroups: well-differentiated lesions (type A, AB, B1 and B2 according WHO) classified as “typical thymoma” and moderately differentiated tumors (type B3 according WHO), classified as “atypical thymoma”. In fact, histology becomes a predictor of relapse when the tumor shows a loss of functional maturity and onset of cytological atypia, features that are distinctive of less differentiated neoplasms [19]. Hence, a simplification of the traditional classification method distinguishing the re-grouped well-differentiated lesions from a single less-differentiated category could lead to a model that better predicts tumor relapses. In the cohort analyzed in our study, the S&M and WHO pathological classifications had an equivalent correlation with tumor stage and DFS, confirming that the main issue may concern the differentiation of moderately and well-differentiated tumors.

Different staging systems for thymic tumors have also been proposed. In our study, a significant correlation of the TNM and M-K staging system with DFS was observed (*p* = 0.002 and *p* = 0.001), showing a substantial equivalence of both staging systems. However, in accordance with previous studies, the analyzed re-staging in the cohort according to the TNM system led to a remarkable shift towards stage I, with a downstaging of Masaoka–Koga stage II and III tumors [20]. This is due to the fact that the current TNM staging system does not differentiate between encapsulated tumors and tumors invading the thymic tissue or mediastinal fat, all classified in stage I. However, it is noteworthy that a recent survey carried out in high-volume thymic-specialized European, American and Asian centers [11] showed that capsular infiltration may still play a prognostic role, and this issue should be specifically dealt with in the definition of the ninth edition of TNM staging system.

In our study, the T parameter was identified as a prognostic factor for DFS. Moreover, a sub-analysis of stage I showed a significant difference in disease-free survival of T1a and T1b tumors. Although based on a relatively limited number of patients, this finding confirms that the stratification of stage I tumors needs to be further analyzed. In fact, in clinical practice, the loss of pathological significance of the involvement of tumor capsule, as for the current TNM classification, means the loss of its therapeutic implication, in particular concerning the indication to adjuvant post-operative radiotherapy [21].

The role of a systematic nodal dissection in the treatment of thymomas is also an issue to be defined. In fact, due to the relatively low incidence of lymphatic metastases (1.8 to 4.2% according to previous studies), the role of lymphadenectomy during surgical treatment of thymic lesions remains a debated subject [22,23,24]. Nevertheless, there is increasing awareness of a possible role of nodal dissection during surgical treatment of thymic tumors, with implications related to the completeness of surgical treatment and accuracy of tumor staging. This issue has been underscored in recent studies that have demonstrated an increase in lymph node metastases (up to 15%) in patients submitted to a systematic nodal dissection [22,25]. Therefore, lymphadenectomy has been proposed by some authors as an essential part of surgical treatment [25,26].

However, since no clear advantage in terms of DFS has been demonstrated after systematic nodal dissection, the optimal extent of nodal dissection is still a matter of debate [22]. A possible option could be a differentiated approach defined according to the potential risk of nodal metastases. This considering that a correlation of the rate of lymph node metastases with the WHO classification and TNM staging system has been reported, with a higher incidence of lymphatic spread in B2-B3 thymomas and in TNM stage III tumors [22].

In the period analyzed in the study, intentional lymph node dissection during thymic surgery was not performed. Nevertheless, patients without lymph nodes metastases in the peri-thymic fat of the surgical specimen (N0) did not have a better DFS than patients with tumors classified as Nx (*p* = 0.86). If confirmed in larger studies these results may confirm that casual dissection of peri-thymic lymph nodes en bloc with the tumor and the mediastinal fat may not be an adequate procedure.

Since a definite histological diagnosis may be obtained only at final pathological examination, preoperative parameters that allow us to identify patients at higher risk of nodal metastases, such as those with less differentiated tumors or more advanced stages of the disease, may be of great importance in the definition of the surgical strategy. In fact, although most of the authors consider extended thymectomy as the standard approach for the treatment of thymic malignancies, others have postulated that in early-stage thymomas without myasthenia a conservative thymomectomy is equal to radical thymectomy in terms of oncological outcome [13,27]. Preoperative CT scan parameters that correlate with stage and histology may therefore be of great importance in the definition of the therapeutic strategy.

In the present study a correlation of radiological features with tumor stage and histology was observed. Atypical thymomas according to the Suster and Moran classification (WHO B3) more often had irregular margins and shape, a wider pericardial contact and a higher incidence of mediastinal fat invasion compared to the typical counterpart. Moreover, TNM stage I tumors did not show pericardial contact or mediastinal fat invasion, in contrast with what was observed in advanced stage lesions. CT scan parameters such as tumor volume and length of pleural contact also influenced disease-free survival.

Only few previous studies aiming at the definition of the radiological characteristics of thymic lesions had shown a correlation of CT scan features with histology and tumor stage. Ozawa et al. [8] evaluated 84 patients with thymoma, and highlighted that “high risk” lesions (B2 and B3 according to WHO) more frequently presented with irregular morphology and pleural spread, while “low risk” thymomas (A, AB and B1) showed regular margins and round shape. Moreover, invasive malignancies (stage III-IV according to M-K) appeared larger, with irregular contours, signs of calcification, necrosis and pleural dissemination. Han et al. [9] confirmed these findings analyzing radiological, clinical and pathological data in 159 patients with thymoma.

The results of the present study confirm that pre-operative CT scan features may allow us to obtain a preoperative definition of thymic lesions in terms of stage, histology and prognosis. All these elements are crucial to direct the patient toward the most appropriate treatment course in order to maximize oncological results.

Major limitations of this study are the relatively small size of the cohort examined and the relative short follow-up, since thymomas are rare and rather indolent pathologies, with may recur even after years after treatment. As disease-free survival was a primary endpoint of the study, both macroscopically and microscopically incomplete (R1 and R2) resections were excluded from the analysis, and therefore factors correlated with incomplete resection were not assessed. Another limitation concerns the fact that lymph node dissection was not routinely performed throughout the study period.

## 5. Conclusions

A correlation of preoperative CT scan parameters with histological and staging classifications and with DFS was observed in surgically treated thymomas. The use of CT parameters may allow a preoperative prognostic stratification of thymomas, improving the definition of the surgical strategy concerning the extent of surgical resection and nodal dissection.

## Figures and Tables

**Figure 1 medicina-57-00010-f001:**
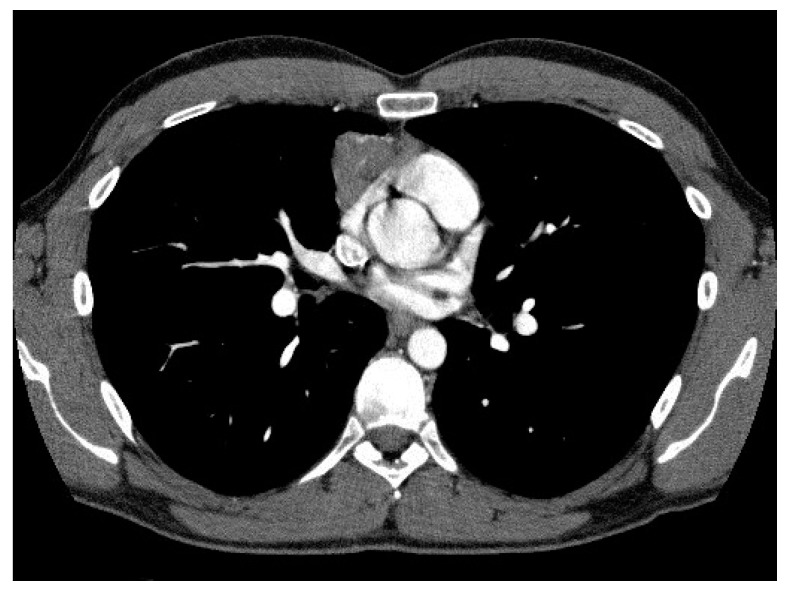
Computed tomography (CT) images showing irregular shape and contours.

**Figure 2 medicina-57-00010-f002:**
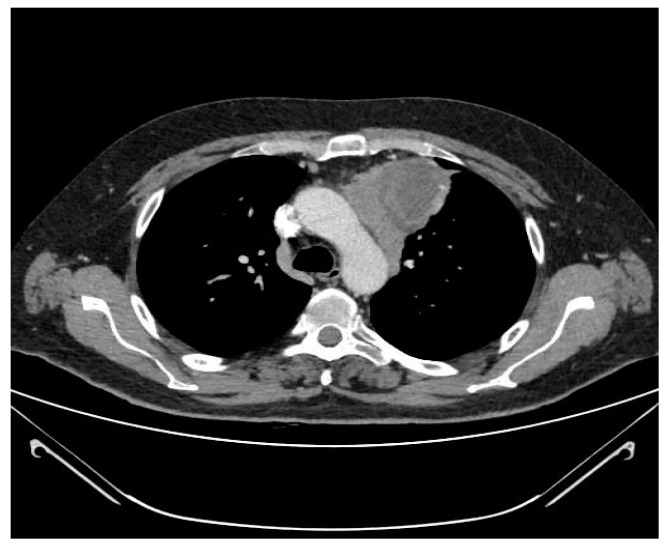
Mediastinal fat invasion.

**Figure 3 medicina-57-00010-f003:**
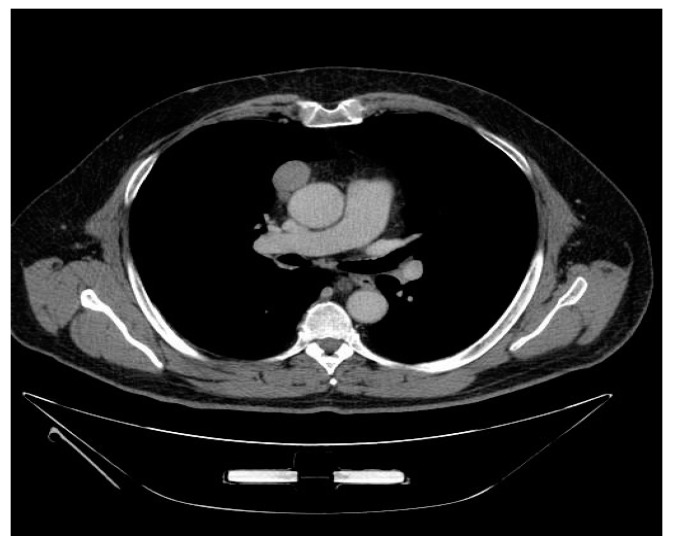
CT image showing oval shape, regular contour thymoma.

**Figure 4 medicina-57-00010-f004:**
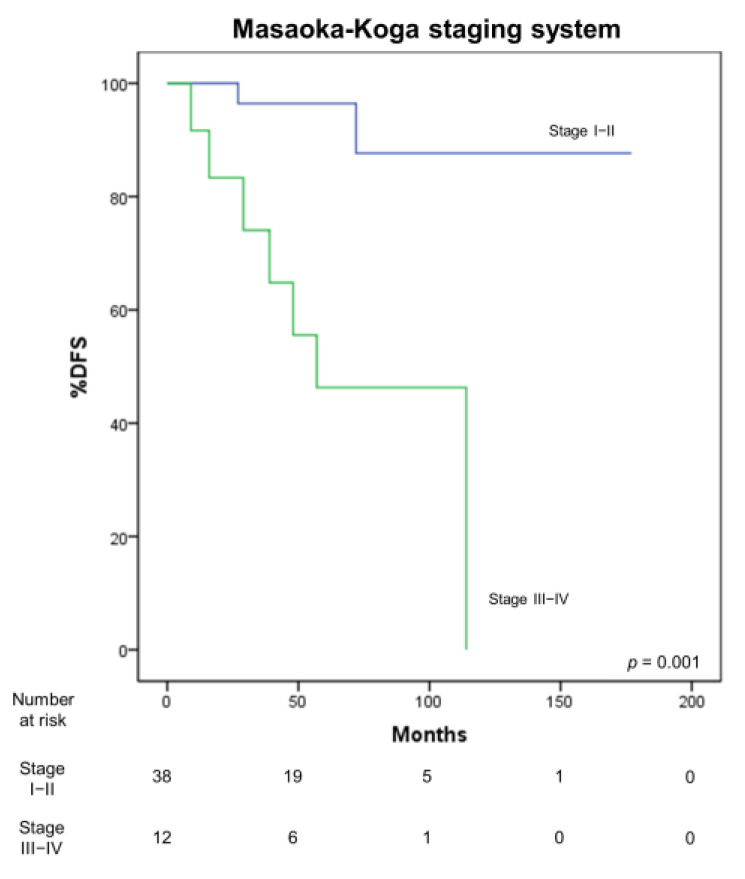
Correlation between disease-free survival (DFS) and Masaoka–Koga staging system.

**Figure 5 medicina-57-00010-f005:**
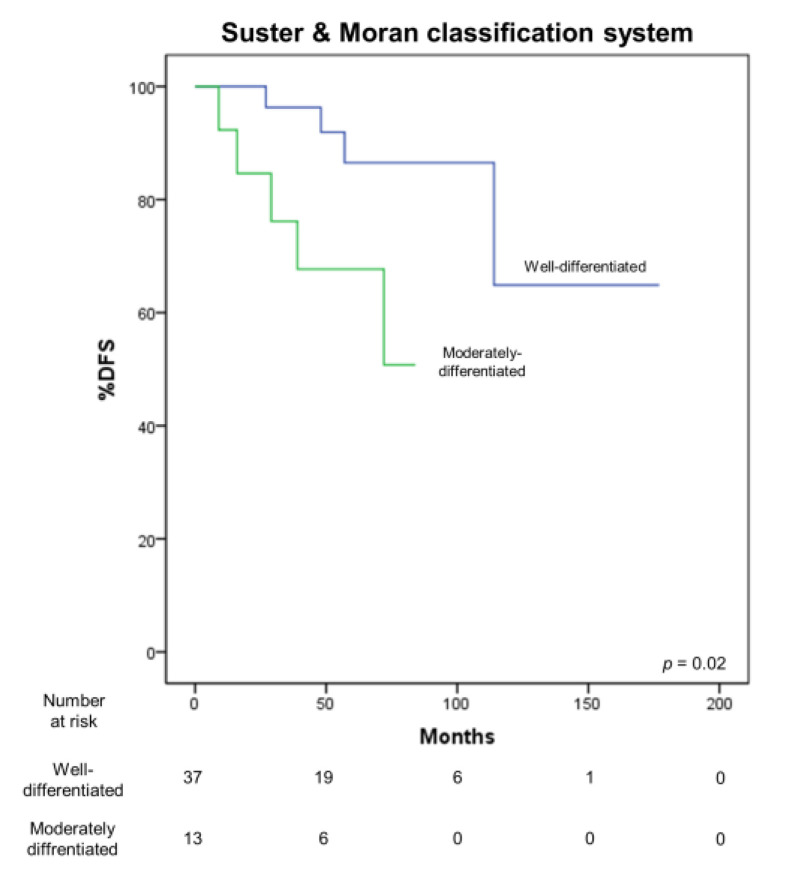
Correlation between DFS and Suster and Moran classification system.

**Figure 6 medicina-57-00010-f006:**
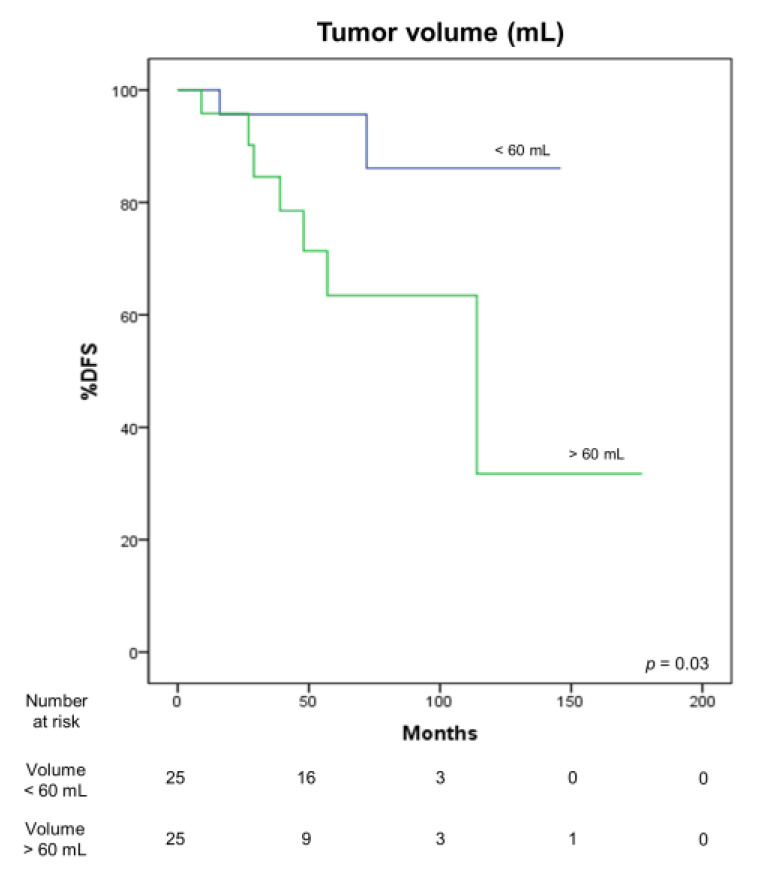
Correlation between DFS and tumor volume.

**Figure 7 medicina-57-00010-f007:**
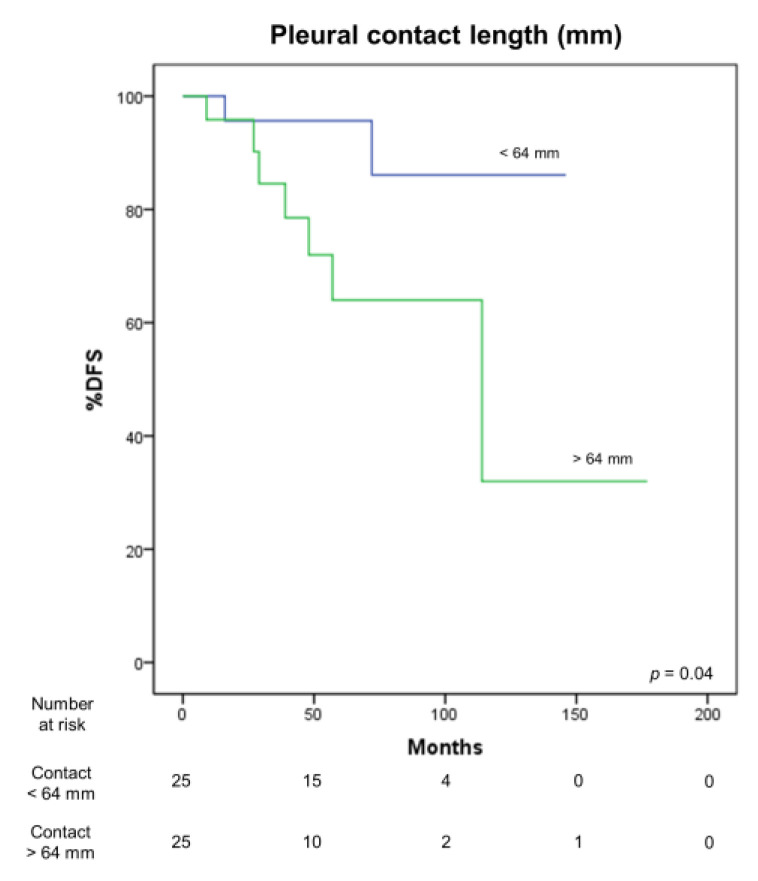
Correlation between DFS and pleural contact length.

**Table 1 medicina-57-00010-t001:** Patients’ characteristics.

Total No.	50
Age (±SD; range)	57.5 ± 12.7; 27–83
Gender (males/female)	27/23
Myasthenia gravis	14 (28%)
Induction treatment	4 (8%)
Surgical access	
Median sternotomy	43 (86%)
Thoracotomy	2 (4%)
Video-assisted thoracic surgery (VATS)	5 (10%)
WHO classification	
A	4 (8%)
AB	14 (28%)
B1	11 (22%)
B2	8 (16%)
B3	13 (26%)
Masaoka stage	
I	13 (26%)
II	25 (50%)
III	10 (20%)
IV	2 (4%)
Adjuvant therapy	30 (60%)

**Table 2 medicina-57-00010-t002:** Classification according to tumor, node and metastases (TNM) and Masaoka–Koga staging systems.

	Masaoka Stage				
	I	II	III	IV	Total No.
TNM stage					
I	13	25	1	0	39 (78%)
II	0	0	2	0	2 (4%)
IIIA	0	0	7	1	8 (16%)
IVA	0	0	0	1	1 (2%)
Total	13 (26%)	25 (50%)	10 (20%)	2 (4%)	

**Table 3 medicina-57-00010-t003:** TNM and WHO histological classifications.

		WHO				
		A	AB	B1	B2	B3
TNM	I	4 (8%)	14 (28%)	11 (22%)	6 (12%)	4 (8%)
	II	0	0	0	0	2 (4%)
	IIIA	0	0	0	2 (4%)	6 (12%)
	IVA	0	0	0	0	1 (2%)

**Table 4 medicina-57-00010-t004:** Correlation between computed tomography (CT) parameters and staging.

		MASAOKA–KOGA				*p*	TNM				*p*
		I	II	III	IV		I	II	III-A	IV-A	
Long axis (median, mm)	<53.5	10 (76.9%)	9 (36%)	5 (50%)	1 (50%)	0.13	20 (51.3%)	0	4 (50%)	1 (100%)	0.39
	>53.5	3 (23.1%)	16 (64%)	5 (50%)	1 (50%)		19 (48.7%)	2 (100%)	4 (50%)	0	
Short axis (median, mm)	<36.5	9 (69.2%)	11 (44%)	4 (40%)	1 (50%)	0.44	20 (51.3%)	0	4 (50%)	1 (100%)	0.39
	>36.5	4 (30.8%)	14 (56%)	6 (60%)	1 (50%)		19 (48.7%)	2 (100%)	4 (50%)	0	
Volume (median, mm^3^)	<60	10 (76,9%)	11 (44%)	3 (30%)	1 (50%)	0.13	21 (53.8%)	0	3 (37.5%)	1 (100%)	0.29
	>60	3 (23.1%)	14 (56%)	7 (70%)	1 (50%)		18 (46.2%)	2 (100%)	5 (62.5%)	0	
Length of pleural contact (median, mm)	<64	10 (76,9%)	11 (44%)	3 (30%)	1 (50%)	0.13	21 (53.8%)	1 (50%)	2 (25%)	1 (100%)	0.36
	>64	3 (23.1%)	14 (56%)	7 (70%)	1 (50%)		18 (46.2%)	1 (50%)	6 (75%)	0	
Shape	Ovalar	13 (100%)	13 (52%)	4 (40%)	0	0.003 *	27 (69.2%)	0	3 (37.5%)	0	0.06
	Irregular	0	12 (48%)	6 (60%)	2 (100%)		12 (30.8%)	2 (100%)	5 (62.5%)	1 (100%)	
Contours	Regular	7 (53.8%)	11 (44%)	1 (10%)	0	0.09	19 (48.7%)	0	0	0	0.034 *
	Irregular	6 (46.2%)	14 (56%)	9 (90%)	2 (100%)		20 (51.3%)	2 (100%)	8 (100%)	1 (100%)	
Necrosis	Yes	5 (38.5%)	11 (44%)	6 (60%)	2 (100%)	0.34	17 (43.6%)	2 (100%)	4 (50%)	1 (100%)	0.31
	No	8 (61.5%)	14 (56%)	4 (40%)	0		22 (56.4%)	0	4 (50%)	0	
Calcifications	Yes	3 (23.1%)	5 (20%)	5 (50%)	0	0.24	9 (23.1%)	1 (50%)	3 (37.5%)	0	0.64
	No	10 (76.9%)	20 (80%)	5 (50%)	2 (100%)		30 (76.9%)	1 (50%)	5 (62.5%)	1 (100%)	
Lymph node enlargements	Yes	1 (7.7%)	5 (20%)	4 (40%)	0	0.51	6 (15.4%)	1 (50%)	3 (37.5%)	0	0.73
	No	12 (92.3%)	20 (80%)	6 (60%)	2 (100%)		33 (84.6%)	1 (50%)	5 (62.5%)	1 (100%)	
Pleural effusion	Yes	2 (15.4%)	1 (4%)	1 (10%)	0	0.63	3 (7.7%)	0	1 (12.5%)	0	0.92
	No	11 (84.6%)	24 (96%)	9 (90%)	2 (100%)		36 (92.3%)	2 (100%)	7 (87.5%)	1 (100%)	
Pericardial effusion	Yes	0	3 (12%)	2 (20%)	0	0.41	3 (7.7%)	0	2 (25%)	0	0.46
	No	13 (100%)	22 (88%)	8 (80%)	2 (100%)		36 (92.3%)	2 (100%)	6 (75%)	1 (100%)	
Invasion of mediastinal fat	Yes	2 (15.4%)	12 (48%)	7 (70%)	2 (100%)	0.022 *	14 (35.9%)	2 (100%)	6 (75%)	1 (100%)	0.05
	No	11 (84.6%)	13 (52%)	3 (30%)	0		25 (64.1%)	0	2 (25%)	0	
Contact with mediastinal vessels	Yes	8 (61.5%)	24 (96%)	10 (100%)	2 (100%)	0.008 *	33 (84.6%)	2 (100%)	8 (100%)	1 (100%)	0.59
	No	5 (38.5%)	1 (4%)	0	0		6 (15.4%)	0	0	0	
Pericardial contact	Yes	7 (53.8%)	21 (84%)	10 (100%)	2 (100%)	0.032 *	29 (74.4%)	2 (100%)	8 (100%)	1 (100%)	0.32
	No	6 (46.2%)	4 (16%)	0	0		10 (25.6%)	0	0	0	
Lung invasion	Yes	1 (7.7%)	0	1 (10%)	0	0.47	1 (2.6%)	1 (50%)	0	0	0.009 *
	No	12 (92.3%)	25 (100%)	9 (90%)	2 (100%)		38 (97.4%)	1 (50%)	8 (100%)	1 (100%)	
Pleural invasion	Yes	1 (7.7%)	1 (4%)	0	0	0.81	2 (5.1%)	0	0	0	0.89
	No	12 (92.3%)	24 (96%)	10 (100%)	2 (100%)		37 (94.9%)	2 (100%)	8 (100%)	1 (100%)	
Contrast enhancement diffusion pattern	Homogeneous	4 (57.1%)	5 (27.8%)	2 (25%)	0	0.42	9 (34.6%)	0	2 (33.3%)	0	0.60
	Inhomogeneous	3 (42.9%)	13 (72.2%)	6 (75%)	1 (100%)		17 (65.4%)	2 (100%)	4 (66.7%)	0	
Laterality	Right	4 (30.8%)	6 (24%)	3 (30%)	0	0.25	11 (28.2%)	1 (50%)	1 (12.5%)	0	0.15
	Left	9 (69.2%)	14 (56%)	3 (30%)	1 (50%)		23 (59%)	0	4 (50%)	0	
	Median	0	5 (20%)	4 (40%)	1 (50%)		5 (12.8%)	1 (50%)	3 (37.5%)	1 (100%)	

* Significant data are marked (*p* < 0.05).

## Data Availability

The data presented in this study are available on request from the corresponding author. The data are not publicly available due to privacy reasons.

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
