# Peer review of "Correlation of Computed Tomography Parameters with Histology, Stage and Prognosis in Surgically Treated Thymomas"

_medicina, 2020, doi:10.3390/medicina57010010_

Round 1

Reviewer 1 Report

Authors retrospectively evaluated CT scan parameters and histology, staging, OS, and DFS in surgically treated thymoma. They included 50 patients. They showed that significant DFS with staging and histology. CT scan parameters correlated with stage, histology, and prognosis. They concluded that CT scan parameters correlate with stage and histology and have a prognostic role in resectable thymoma. Data are clearly presented.

I have the following concerns.

  1. In figure 1, correlation between DFS and histology was presented. Authors should present other factors, including tumor volume, length of pleural contact, and Masaoka stage.
  2. In figure 1, authors should show number at risk at each time point.

Author Response

Re: ‘Correlation of computed tomography parameters with histology, stage and prognosis in surgically-treated thymomas’

Point-by point response to the Reviewers’ comments

We thank the Editors and Reviewers for their significant contribution in improving the text.

The changes have been highlighted in red in the revised version of the manuscript

Reply to Reviewer 1 comments:

Comment 1: In figure 1, correlation between DFS and histology was presented. Authors should present other factors, including tumor volume, length of pleural contact, and Masaoka stage

Reply: Figures presenting the correlation of disease free survival with Masaoka stage, tumor volume and length of pleural contact have been added to the manuscript as suggested (See Figures 4, 6 and 7)

Comment 2: In figure 1, authors should show number at risk at each time point.

Reply: The numbers at risk at each time point have been added to the figures as advised

Yours sincerely,

Angelo Carretta, MD

Reviewer 2 Report

General comments:

This retrospective study evaluates the correlation between CT features and histopathology, stage and prognosis in 50 patients surgically resected thymomas. Overall, the study is clearly written and interesting. However, since this is mainly a radiological article, radiological images should be provided. It is suggested to add at least one example of tumors with aggressive CT features and worse prognosis, and a tumor with lower aggressiveness. This will be very helpful for future readers in the preoperative assessment of thymomas.

Specific comments:

Abstract: Adequate.

Introduction:

-Please detail how CT features could change patients’ managements in the preoperative setting.

Materials and Methods:

-Study design: Why did the Authors excluded patients with R1 resection? Exclusion of R2 resection could be understandable, but it could also be interesting to assess radiological features that could be correlated with R1 resection and worse prognosis.

-Please add a brief paragraph with detailed CT scanning protocol (i.e. contrast-enhanced phases with scanning timing) and parameters in the Methods.

-CT scan features: “CT scans were reviewed by two radiologists”. Where the radiologists blinded to the histopathological diagnosis at the time of imaging review?

-Table 1 reports the CT scan features in the whole cohort. This table looks more appropriate in the Results sections, or even better, in the supplementary materials.

Results:

-Supplementary table 1 looks really important for the interpretation of the results. It is suggested to provide this table in the main text. Instead Table 1 could be moved in the supplementary since this is not representative of correlation of CT scan parameters, histology and staging, which is the main purpose of this study.

-Please provide CT images with examples of thymomas with irregular shape and contours, mediastinal fat invasion or pericardial contact.

Discussion:

-Please discuss limitations of the present study in a dedicated paragraph before the conclusions. Discuss the exclusion of R1 resections that could lead to significant selection bias.

Author Response

Re: ‘Correlation of computed tomography parameters with histology, stage and prognosis in surgically-treated thymomas’

Point-by point response to the Reviewers’ comments

We thank the Editors and Reviewers for their significant contribution in improving the text.

The changes have been highlighted in red in the revised version of the manuscript

Reply to Reviewer 2 comments:

Comment 1: It is suggested to add at least one example of tumors with aggressive CT features and worse prognosis, and a tumor with lower aggressiveness

Reply: CT scan images of tumors with aggressive CT features and worse prognosis and of a tumor with low aggressive features have been added to the manuscript as suggested (see Figures 1, 2 and 3).

Introduction

Comment 2: Please detail how CT features could change patients’ management in the preoperative setting

Reply: The potential impact of CT features on patients’ management in the preoperative setting has been added to the Introduction section (see line 57).

Materials and Methods

Comment 3: Study design: Why did the Authors excluded patients with R1 resection? Exclusion of R2 resection could be understandable, but it could also be interesting to assess radiological features that could be correlated with R1 resection and worse prognosis.

Reply: Since disease free survival was identified as a primary endpoint of the study, incomplete resections (R1 and R2) were excluded from the analysis. This point has been specified in the Methods section in line 69. The potential limitations related to this choice have been reported in the Discussion section (See line 341).

Comment 4: Please add a brief paragraph with detailed CT scanning protocol (i.e. contrast-enhanced phases with scanning timing) and parameters in the Methods.

Reply: A paragraph with detailed CT scan scanning protocol and parameters has been added to the methods section (See line 93)

Comment 5: CT scan features: “CT scans were reviewed by two radiologists”. Where the radiologists blinded to the histopathological diagnosis at the time of imaging review?

Reply: Both radiologists were blinded to the histopathological diagnosis of the tumors. This point has been specified in the Methods section (See line 100).

Comment 6: Table 1 reports the CT scan features in the whole cohort. This table looks more appropriate in the Results sections, or even better, in the supplementary materials.

Reply: Table 1 has been moved to the supplementary materials as suggested

Results

Comment 7: Supplementary table 1 looks really important for the interpretation of the results. It is suggested to provide this table in the main text. Instead Table 1 could be moved in the supplementary since this is not representative of correlation of CT scan parameters, histology and staging, which is the main purpose of this study.

Reply: Supplementary Table 1 has been moved to the main text as suggested, and original Table 1 has been moved to the supplementary material as Supplementary Table 1

Comment 8: Please provide CT images with examples of thymomas with irregular shape and contours, mediastinal fat invasion or pericardial contact.

Reply: CT scan images of thymomas with irregular shape and contours and mediastinal fat invasion have been added (See Figures 1 and 2)

Discussion

Comment 9: Please discuss limitations of the present study in a dedicated paragraph before the conclusions. Discuss the exclusion of R1 resections that could lead to significant selection bias.

Reply: A paragraph analyzing the limitations of the study and in particular the potential implications related to the exclusion of R1 resections has been added to the Discussion section (See line 339)

Yours sincerely,

Angelo Carretta, MD